# Reservoir of Antibiotic Residues and Resistant Coagulase Negative Staphylococci in a Healthy Population in the Greater Accra Region, Ghana

**DOI:** 10.3390/antibiotics11010119

**Published:** 2022-01-17

**Authors:** Samuel Oppong Bekoe, Sophie Hane-Weijman, Sofie Louise Trads, Emmanuel Orman, Japheth Opintan, Martin Hansen, Niels Frimodt-Møller, Bjarne Styrishave

**Affiliations:** 1Department of Pharmaceutical Chemistry, Faculty of Pharmacy and Pharmaceutical Sciences, Kwame Nkrumah University of Science and Technology, Kumasi, Ghana; 2Toxicology and Drug Metabolism Group, Department of Pharmacy, Faculty of Health and Medical Sciences, University of Copenhagen, Universitetsparken 2, 2100 Copenhagen, Denmark; sophie.haneweijman@gmail.com (S.H.-W.); sofie.trads@gmail.com (S.L.T.); martin.hansen@envs.au.dk (M.H.); bjarne.styrishave@sund.ku.dk (B.S.); 3Department of Pharmaceutical Chemistry, School of Pharmacy, University of Health and Allied Sciences, Ho, Ghana; eorman@uhas.edu.gh; 4Department of Medical Microbiology, University of Ghana Medical School, Accra, Ghana; jaopintan@chs.ug.edu.gh; 5Department of Environmental Sciences—Environmental Chemistry and Toxicology, Aarhus University, Frederiksborgvej 399, 4000 Roskilde, Denmark; 6Department of Clinical Microbiology, Rigshospitalet, Blegdamsvej 9, 2100 Copenhagen, Denmark; niels.frimodt-moeller@regionh.dk

**Keywords:** antimicrobial resistance, resistant CoNS, healthy individuals, LC-MS/MS, antimicrobial susceptibility test

## Abstract

Antimicrobial resistance threatens infectious disease management outcomes, especially in developing countries. In this study, the occurrence of resistant coagulase-negative staphylococci (rCoNS) and antibiotic residues in urine samples of 401 healthy individuals from Korle-Gonno (KG) and Dodowa (DDW) in Ghana was investigated. MALDI-ToF/MS with gram-staining techniques detected and identified the CoNS. SPE-LC-MS/MS detected and quantified nine commonly used antibiotics in the samples. The results showed 63 CoNS isolates detected in 47 (12%) samples, with *S. haemolyticus* (78%) and *S. epidermidis* (8%) being predominant. Most of the isolates (95%) were resistant to at least one antibiotic, with the highest resistance observed against sulphamethoxazole (87%). Resistance profiles in samples from DDW and KG were largely comparable, but with some differences. For instance, DDW isolates were more resistant to gentamicin (*p* = 0.0244), trimethoprim (*p* = 0.0045), and cefoxitin (*p* = 0.0078), whereas KG isolates were more resistant to erythromycin (*p* = 0.0356). Although the volunteers had not knowingly consumed antibiotics two weeks before sampling, antibiotic residues, ranging between 1.44–17000 ng mL^−1^ were identified in 22% of urine samples. Samples with antibiotic residues were likely to also contain rCoNS (89%). The most frequent antibiotics detected were tetracycline (63%) and ciprofloxacin (54%). Healthy individuals could thus be reservoirs of antibiotic residues and rCoNS at the community level.

## 1. Introduction

Infectious diseases remain among the leading contributors to sickness and early mortality in populations of third-world nations. The global burden of communicable diseases, including infections, though declining over the past two decades, remains significant [1,2]. Notwithstanding antibiotic usage, infectious bacterial diseases continue to account for significant global mortalities, killing approximately 3.8 million children under the age of five each year [3].

Antimicrobial resistance (AMR) is becoming an increasingly frequent global threat in hospital and community settings. AMR poses a major challenge to local, national, and global public health, costing human lives and livelihoods. Data available show that the developing world record alarming rates of resistance development against several kinds of pathogens to commonly prescribed antibiotics including ampicillin, nalidixic acid, and fluoroquinolones [4,5,6]. Zaidi et al. [7] reported that rates of hospital-acquired infections in developing countries occur 3–20 times more often than in developed countries, and that 70% of these infections cannot be treated by the WHO-recommended regimens due to AMR. 

Several factors, including exposure to sub-therapeutic antimicrobial doses, substandard medicines for treatment, antibiotic residues in plant- and animal-based food products, and adulterated herbal medicines, have been proposed to account for high AMR frequency [8,9,10]. 

Coagulase negative staphylococci (CoNS) constitute part of the normal human microbiota and have been shown to reduce skin colonization by harmful pathogens [11,12]. Until recently, their presence in hospitalized patients had been considered as harmless microbial contaminants [13]. However, some reports show that CoNS-related infections pose challenges in healthcare delivery [14]. For instance, they are known to cause infections such as pneumonia, omphalitis, and abscesses in neonates, and endophthalmitis and keratitis among older children [15]. Their presence is also associated with the frequent cause of peritonitis in patients who undergo peritoneal dialysis [16]. They are also reported in conditions such as urinary tract infections, endocarditis, and central nervous system shunt infections, among others [17]. With recent publications on their resistance developments to commonly used antibiotics [18,19], infections from these organisms are expected to be more challenging. Moreover, most CoNS-related infections are known to be hospital-acquired [16], and so are not expected to be detected at significant levels within the community. Their presence within a population, especially the resistant CoNS (rCoNS), may indicate the spread of resistant microorganisms among individuals, and such information is deemed critical to the control of AMR. This study was conducted to establish antibiotic residues and rCoNS presence within a healthy population, and the correlation of their occurrence at the community level. Data on such occurrences are therefore needed to understand the complex connections that contribute to the rapid spread of AMR. 

## 2. Results

### 2.1. Microbiological Assay

A total of 63 isolates of CoNS were detected in 47 of the 401 urine samples analyzed (Appendix A). The most isolated CoNS was *S. haemolyticus* (77.8%, *n* = 49/63), and its prevalence was significantly higher (*p* = 0.0016) than *S. epidermidis* (7.9%, *n* = 5/63) and other CoNS strains (14.3%, *n* = 9/63) (Figure 1). A total of eight different CoNS species were isolated (Table 1). Most of the CoNS isolated (95.2%, *n* = 60/63) were resistant to at least one of the investigated antibiotics.

The resistance observed in isolates from both DDW (97.3%, *n* = 36/37) and KG (92.3%, *n* = 24/26) were comparable (*p* = 0.1696). In both communities, the highest resistance was shown against sulfamethoxazole (DDW—86.5%; KG—88.5%), followed by benzylpenicillin (DDW—91.9%; KG—76.9%), tetracycline (DDW—68%; KG—66%), trimethoprim (DDW—73.0%; KG—46.2%), and chloramphenicol (DDW—62.2%; KG—34.6%) (Figure 2A). Significantly higher resistance was observed in isolates from DDW against gentamicin (*p* = 0.0244), trimethoprim (*p* = 0.0045), and cefoxitin (*p* = 0.0078) than in isolates from KG, while resistance was higher in isolates from KG against erythromycin than from DDW (*p* = 0.0356) (Figure 2A). There was, however, no resistance found to clindamycin and vancomycin in isolates from KG, and relatively little was observed in isolates from DDW (5.4% and 2.7%, respectively). *S. epidermidis* showed a higher percentage of resistance to the majority of the test antibiotics (57.1%, *n* = 8/14), especially against sulfamethoxazole (100%) and benzylpenicillin (100%), followed by *S. haemolyticus* and the other CoNS strains (Figure 2B). There was, however, no significant difference in resistance to individual antibiotics when comparing CoNS species (*p* > 0.05).

Multi-drug resistance (MDR) was also investigated. Table 1 shows the number of CoNS isolates displaying MDR, as defined by expressing resistance to four or more antibiotics from separate groups. In total, 69.8% (*n* = 44/63) of the CoNS isolates showed resistance to ≥ four antibiotics. Additionally, MDR *S. haemolyticus* was the predominant MDR CoNS isolate (71.4%, *n* = 35/44) detected, and was more prevalent in samples from DDW (68.6%, *n* = 24/35) than from KG (31.4%, *n* = 11/35) (*p* = 0.0273).

### 2.2. Pharmaceutical Analysis

The validation results on the method developed for the detection and quantitation of the antibiotics in the urine samples are shown in Appendix A. Briefly, the validation considered analytes in the concentration range of 500–5000 ng mL^−1^, with limits of detection and quantitation (LOD & LOQ) ranging between 310–1180 ng mL^−1^ and 940–3570 ng mL^−1^, respectively. The SPE absolute recoveries were observed to range between 85–107% for the analytes investigated. From the analysis, there were seven distinct types of antibiotics detected in 22.2% (*n* = 89/401) of the urine samples analyzed (Figure 3). Out of the urine samples containing CoNS, 89.4% (*n* = 42/47) had also recorded the presence of antibiotic residues (Appendix A). It was evident that samples containing antibiotics were more likely to also contain rCoNS (*p* = 0.0002). Furthermore, 59.6% of samples from DDW showed the presence of antibiotics, as compared to 40.4% of samples from KG (Appendix A). The most frequently occurring antibiotics in the urine samples were tetracycline (62.9%, *n* = 56/89) and ciprofloxacin (53.9%, *n* = 48/89) (Appendix A).

Out of the 89 samples showing the presence of antibiotics, 75 of them (84.3%) had quantifiable levels of the antibiotics: 36 (48.0%) from KG and 39 (52.0%) from DDW. A considerable number of these samples contained quantifiable levels of one (57.3%, *n* = 43), two (30.7%, *n* = 23), three (8.0%, *n* = 6), four (2.7%, *n* = 2), and five (1.3%, *n* = 1) of the antibiotics. The general observation was that a similar number of samples from both KG and DDW contained more than one quantifiable residual level of antibiotics (50.0%, *n* = 16). The analysis further showed that the residual levels of the antibiotics were significantly different from each other (*H* = 27.02, *p* = 0.0001), with metronidazole (*n* = 3) recording the highest median concentration (167 ng mL^−1^), followed by tetracycline (*n* = 41, 26.7 ng mL^−1^), and ciprofloxacin (*n* = 48, 14.6 ng mL^−1^). Trimethoprim (*n* = 9), on the other hand, recorded the least median concentration (2.91 ng mL^−1^). The concentrations of antibiotics determined are summarized in Figure 4.

## 3. Discussion

### 3.1. Identified CoNS Isolates and Their Resistance to Antibiotics

The antibiotics assessed represented one resistance type or marker each. However, resistance towards cefoxitin in staphylococci was a marker for methicillin resistance, which would also result in resistance towards benzylpenicillin, while resistance towards benzylpenicillin in cefoxitin-susceptible strains represented the presence of a penicillinase. Both were counted as separate markers for resistance, based on the hypothesis that all methicillin-rCoNS also carried genes for penicillinase production. The detection of CoNS isolates with resistance to the investigated antibiotics from participants not on therapeutic medicines and assumed to be otherwise healthy is a worrying observation.

The detection (Figure 1), coupled with their demonstrated resistance profiles (Figure 2; Table 1), supports the exposure (unknowingly and unintentionally) of healthy individuals to antimicrobial agents, from sources such as plant- and animal-based food products, as well as adulterated herbal medicines [9,20,21]. This could lead to AMR development by organisms, hitherto considered harmless and beneficial [22]. For example, *S. epidermidis* and S haemolyticus, known to be a commensal CoNS species on the human skin [23], demonstrated varying levels of resistance against the test antibiotics in this study. *S. epidermidis* was resistant against sulfamethoxazole (100%), trimethoprim (80%), tetracycline (80%), and chloramphenicol (60%) (Figure 2B). Similar observations were made by De Allori et al. on *S. epidermidis* in in-patients [24]. In the same study, S haemolyticus also showed resistance against trimethoprim-sulfamethoxazole (29%), tetracycline (43%), and chloramphenicol (36%) [24]. These outcomes support the evidence that both patients and healthy individuals could be sources of rCoNS, and their prevalence in the general population may contribute to AMR. 

The observed resistance profiles, as shown among healthy individuals in the current study (Figure 2A,B), were also reported by Newman et al. [25] and Lerbech et al. [14] among patients in different parts of Ghana. The outcomes of these two studies significantly correlate with that of the present study, in that, four of the top five antibiotics with a high prevalence of resistance against them in each of these studies are similar, namely sulfamethoxazole, trimethoprim, tetracycline, and chloramphenicol. This pattern indicates that irrespective of the different settings and scope of the three studies, the reported resistance may be widespread in the general population. The situation looks dire especially in the management of infectious diseases in rural Ghana, where the population engages in self-medication with poor-quality and cheaper options of antibiotics [8,26]. The preference for cheaper broad-spectrum antibiotics in the population [26] could explain the high detection of tetracycline and ciprofloxacin (Figure 3). 

Vancomycin resistance in one of the isolates, (MIC > 4 mg L^−1^) was, however, not validated by further testing, and could therefore be considered as an artifact. This observation supports the outcomes of a previous study by Lerbech et al. on samples from patients, where there was no record of resistance against the same drug [14]. Vancomycin is considered the last-resort, drug of choice, for the treatment of many infectious diseases, including those caused by staphylococci [27]. 

### 3.2. Antibiotics in Urine Samples

Tetracycline and ciprofloxacin constituted the antibiotics most frequently detected in the urine samples (Appendix A). Their detection indicates that the participants were exposed chronically to these antibiotics from alternate sources. 

One of such alternate sources could be adulterated herbal medicines, where manufacturers seek to augment the antimicrobial efficacy of their products with either low or too high concentrations of known antibiotics [21,28,29]. The use of such adulterated herbal medicines could result in lower concentrations of antibiotics being detected. Although no scientific report of such detection has been reported in Ghana, the recall of herbal products from the Ghanaian market by the Food and Drugs Authority, Ghana, reported to contain gentamycin, streptomycin, sulphathiazole, dexamethasone, diazepam, and chloramphenicol, among others [30], show that herbal medicine adulteration with antibiotics is a common practice. It is worthy to note that some participants in the current study had patronized herbal medicines (Appendix A) (data not reported). 

Animal and fish products (Appendix A) could also be considered as potential sources of these antibiotics as their use in managing or preventing infections in the veterinary sector leads to their presence in the animal food chain [9,20,31,32,33]. 

Concerns with the indiscriminate use of antibiotics in industrial aquaculture have led to the establishment of maximum residue limits of some antibiotic residues by the European Union and other international regulatory bodies [34]. In Ghana, Agoba et al. reported the addition of tetracycline and chloramphenicol to fish feed by some fish farmers in the Ashanti region [35]. This practice may lead to the presence of these antibiotics in fish products. For example, a preliminary study by Donkor et al. had demonstrated the presence of 16 different antibiotic residues in 7% of Nile Tilapia fish sampled from four different markets in Ghana [36]. Residues of various antibiotics belonging to various classes have also been reported in egg, fish, milk, cheese, yogurt, and leafy vegetables in Ghana [9,20,37]. 

These antibiotics are often present at sub-therapeutic levels in these food products, and healthy individuals are unknowingly exposed to them, hence, the levels of the antibiotics detected in the current study (Appendix A), being lower than their minimum inhibitory concentrations which for most microbes, are in the µg mL^−1^ range [38]. Chronic exposure to sub-therapeutic levels of antibiotics may select for resistance [39] which may in turn render therapeutic treatments ineffective. Consequently, unintentional chronic exposure of the general public to sub-therapeutic levels of antibiotics from indirect sources such as food, water, and herbal medicine may challenge public health. It was established in the present study that urine samples containing antibiotic residues were more likely to also contain rCoNS (89.4%; *p* = 0.0002). It may be possible, however, that the concentrations of some of the antibiotics (for example, tetracyclines) could have been higher in the consumed food products and show up in lesser amounts in the urine because they are either metabolized extensively or excreted in relatively higher concentrations in feces than in urine. In either case, the occurrence is deemed unacceptable and calls for proper control of such medicines. 

This study supports earlier findings that show that the Ghanaian population is exposed to antibiotics from various sources other than prescribed medicines, and such a trend could influence resistance formation in bacteria and ultimately, public health. 

## 4. Materials and Methods

### 4.1. Sampling Sites

A total of 401 urine samples were collected from healthy individuals in Korle-Gonno (KG) (*n* = 200) and Dodowa (DDW) (*n* = 201), both in the southern part of Ghana. KG is an urban area in the neighborhood of the Korle-Bu Teaching Hospital (KBTH) in Accra, and DDW is a peri-urban town, located 100 km from Accra within the Dangbe-West District. Both locations were chosen based on the outcomes of a previous study [14]. 

### 4.2. Participant Screening and Sample Collection

In DDW, participants were selected systematically, using a local database that randomly selected households throughout the community. In KG, participants were selected randomly, but not systematically. At both sites, 25% of healthy volunteers were selected from each of four quadrants of the communities, based on a protocol from the Ghana Health Service. The distribution of male to female samples was 40:60. The ages of participants ranged between 5–83 years. Persons with records of antibiotics consumption two weeks before the study were excluded. Additionally, any other medication a person had consumed was documented (Appendix A). Consent was sought from all participants included in the study (Appendix A). Urine samples were provided by participants in sterile 120 mL cups and stored at −18 °C before microbiological and antibiotic analyses. 

### 4.3. Microbiological Assay

#### 4.3.1. Urine Sample Screening, and CoNS Isolation and Identification

Microbiological analyses were conducted on the samples from KG and DDW. 10 µL of urine samples were inoculated on Discovery agar plates and incubated at 37 °C for 24 h to allow for different bacterial growths identification using Flexicult SSI-Urinary Kits [40]. Staphylococcal colonies with distinct morphologies were isolated for further testing on MacConkey plates (prepared in–house). Staphylococcal identification was performed using microscopy (Olympus CX31, Olympus Danmark A/S, Ballerup, Denmark) and gram staining (Becton, Dickinson & Co., Kongens Lyngby, Denmark), and, in some cases, a catalyst reaction test, following the European Urinalysis Guidelines and American Society for Microbiology’s Manual of Clinical Microbiology [41]. Identified CoNS were cultivated at 37 °C for 24 h in Mueller–Hinton agar, isolated, and confirmed with MALDI-TOF/MS (Bruker Biotyper system, Microflex LT/SH MS, Roskilde, Denmark). Periodic monitoring of its accuracy, sensitivity, and specificity were ensured. Using the FlexiControl software and Biotyper real-time classification (RTC), a score value greater than 1.8 was accepted as identified. CoNS were re-plated and isolated on 5% blood agar, and colony matter was placed directly on the target plate (MSP 96 target polished steel BC Microscout Target) for analysis, in replicates for accuracy, and covered with a 1 μL of the matrix solution. *Staphylococcus aureus* ATCC 25923 was used as a control strain.

#### 4.3.2. Antimicrobial Susceptibility Testing

The antimicrobial susceptibility test was conducted following protocols from the European Committee on Antimicrobial Susceptibility Testing (EUCAST) on the identified CoNS isolates with standard antibiotics (selected based on recommendations by the Clinical and Laboratory Standards Institute (CLSI) and EUCAST guidelines). Susceptibility was determined based on EUCAST breakpoint data (Appendix A) corresponding to coagulase-negative staphylococci [42] on each isolate growth. *Staphylococcus aureus* ATCC 25923 was used as a control strain.

### 4.4. Analysis of Antibiotic Residues in Urine Samples

#### 4.4.1. Reagents and Chemicals

The chemicals included acetonitrile (ACN), and methanol (MeOH). Reference drugs were ampicillin, amoxicillin, cefuroxime, ciprofloxacin, erythromycin, metronidazole, sulfamethoxazole, tetracycline, and trimethoprim. Internal standards (IS) included d4-sulphamethoxazole, d8-ciprofloxacin, and d3-trimethoprim, and the additives were formic acid (FA) (98–100%) and triethylamine (TEA) (≥99%). They were all of the required grades and purities, respectively. 

#### 4.4.2. Preparation of Standard Solutions and Mobile Phases

Stock solutions of IS mixture containing d8-ciprofloxacin, d4-sulphamethoxazole, and d3-trimethoprim of concentration 10 μg mL^−1^ each, and individual antibiotics at 100 μg mL^−1^ were prepared. Working concentrations at 10 μg mL^−1^ for a mixture of IS and standard antibiotics were then analyzed using a solvent system consisting of mobile phase A (998 mL Milli-Q water and 2 mL ACN and mobile phase B (50 mL Milli-Q water and 950 mL ACN). Both solvent systems contained additives (100 μL each for FA and TEA). All solutions were stored at −18 °C before analyses. 

#### 4.4.3. Sample Preparation and Clean-Up

Sample preparation and clean-up before analyses were ensured by solid-phase extraction (SPE) with Oasis^®^ HLB 6 cm^3^ 200 mg (30 μm) cartridges from Waters (Milford, MA, USA). IS mixture (100 μL of 10 μg mL^−1^) was added to each urine sample before freezing. All protocols employed were optimized and validated as reported in earlier studies [14,43]. Positive controls comprising 100 μL of 10 μg mL^−1^ deuterated standards and 100 mL distilled water were analyzed intermittently to assure the quality of sample clean-up data.

Elution of the SPE cartridges was performed with 5 mL mobile phase A: mobile phase B (10:90, *v/v*), followed by 3 mL MeOH: water (20:80, *v/v*), and subsequently dried completely under a gentle flow of nitrogen gas at room temperature. Dried extracts were then re-dissolved with 200 μL of mobile phase A: mobile phase B (10:90, *v/v*) and transferred into 200 μL flat cap HPLC vials (Chromacol) for analysis using the HPLC-MS/MS system. 

SPE absolute recovery was investigated at a 1000 ng mL^−1^ concentration for the nine compounds under study.

#### 4.4.4. Chromatographic Conditions

The chromatographic system employed was as reported in the literature [14,43,44]. The column used was an XTerra^®^ MS C18 column (100 mm × 2.1 mm × 3.5μm) and the guard column XTerra^®^ MS RP18 column (2.0 mm × 3.0 mm × 3.5μm). Both columns came from Waters (Milford, MA, USA). A binary gradient elution program consisting of the two mobile phase solutions A and B was: 0–5 min: 99% A and 1% B; 5–28 min: 99% to 55% A and 1% to 45% B; 28–32 min: 55% to 5% A and 45% to 95% B; and finally, 32–45 min: 5% to 99% A and 95% to 1% B. An injection volume of 5 μL and a flow rate of 250 μL min^−1^ at a column temperature of 30 °C was used in the LC-MS/MS analysis. The mass spectral analyses were performed with electrospray ionization in both negative and positive ionization modes.

#### 4.4.5. Development and Validation of the LC-MS/MS Method

The LC-MS/MS method was developed and optimized by monitoring the retention times of the antibiotics (standard solutions) and adjusting dwell times to enhance peak sensitivities. The optimized instrumental parameters employed (Appendix A) were then used to separate, identify, and quantify the antibiotics in both negative and positive ionization modes. Figure 5 shows the final chromatogram of the LC-MS/MS method. The method was validated as per the guidelines of the International Council for Harmonisation [45] (Appendix A).

The urine samples were analyzed for the presence of amoxicillin, ampicillin, cefuroxime, ciprofloxacin, erythromycin, metronidazole, sulfamethoxazole, tetracycline, and trimethoprim, using the validated method. 

### 4.5. Statistical Analysis

All chromatographic data were analyzed using MS analysis software (Analyst v1.4) and Microsoft Excel (2008). GraphPad Prism 6 (version 6.01) was used for Fischer’s Exact, two-way ANOVA, and the Kruskal–Wallis tests. 

## 5. Conclusions

The application of advanced mass spectrometric techniques in the microbiological and pharmaceutical analysis of urine samples has established the presence of resistant strains of CoNS and antibiotic residues in healthy individuals. The study has also established that samples with antibiotic residues were likely to also contain rCoNS. The emerging phenomenon signifies that the general populace could serve as reservoirs of both resistant strains of CoNS as well as residual levels of antibiotics from unintentional sources.

## Figures and Tables

**Figure 1 antibiotics-11-00119-f001:**
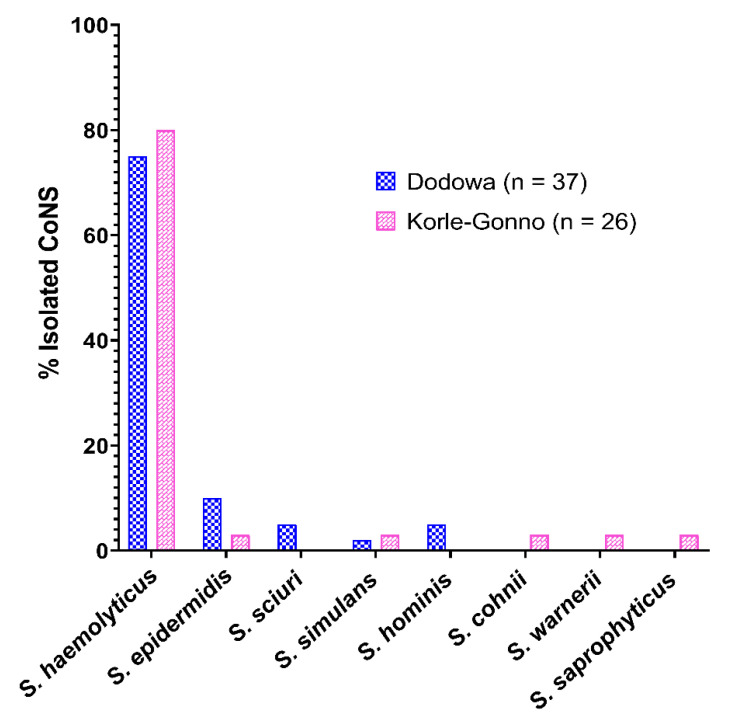
Prevalence of identified CoNS species from the different collection sites. Data expressed as proportions of the different CoNS species observed. Their differences were tested with Chi-square test and were found to be significantly different (*χ*^2^ = 23.11, *df* = 7, *p* = 0.0016).

**Figure 2 antibiotics-11-00119-f002:**
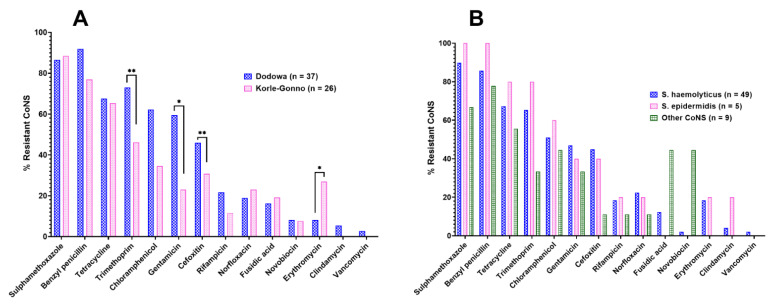
Resistance of isolated CoNS species to test antibiotics. (**A**) Distribution of resistance of isolates from the two communities. (**B**) Breakdown of resistance of the different CoNS species to the test antibiotics. Statistical analysis by two-way ANOVA with Sidak’s post-test showed significant differences in some of the proportions of resistant CoNS against some antibiotics from the two communities: * *p* < 0.05 & ** *p* < 0.01.

**Figure 3 antibiotics-11-00119-f003:**
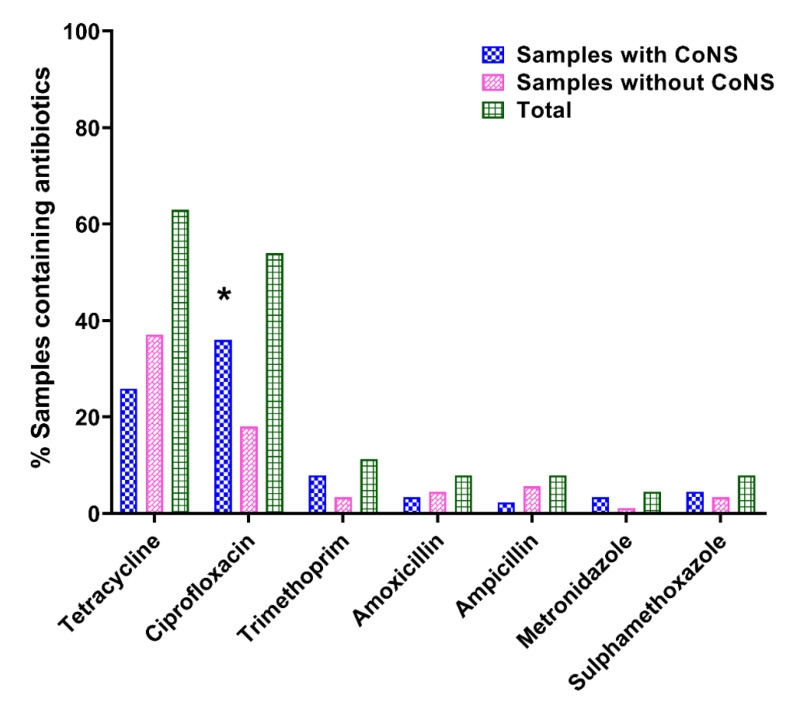
Presence of individual antibiotics in urine samples of participants, shown by the presence of CoNS. Statistical analysis by two-way ANOVA with Sidak’s post-test showed a significant difference in the samples with and without CoNS containing ciprofloxacin: * *p* < 0.05.

**Figure 4 antibiotics-11-00119-f004:**
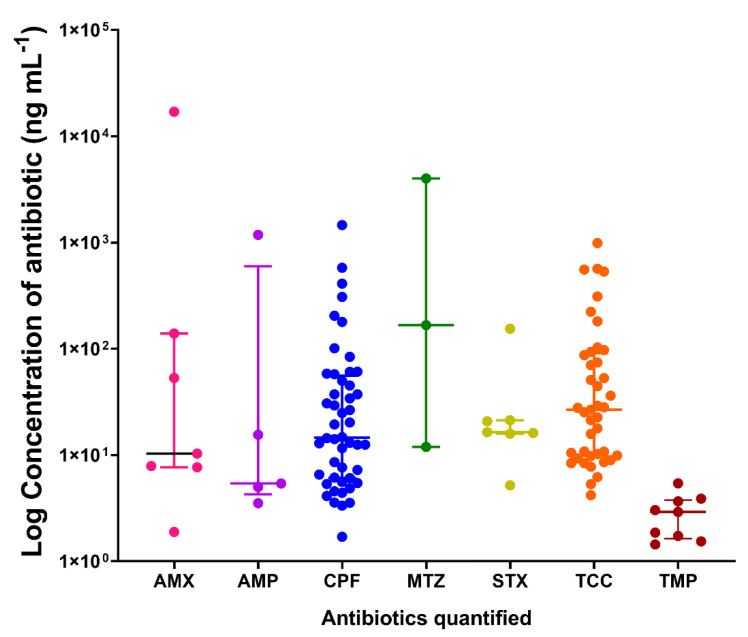
A scatter plot of the residual concentrations of antibiotics present in urine samples. Data shown are the median concentrations of the antibiotics with their interquartile ranges. AMX—Amoxicillin; AMP—Ampicillin; CPF—Ciprofloxacin; MTZ—Metronidazole; STX—Sulfamethoxazole; TCC—Tetracycline, and TMP—Trimethoprim.

**Figure 5 antibiotics-11-00119-f005:**
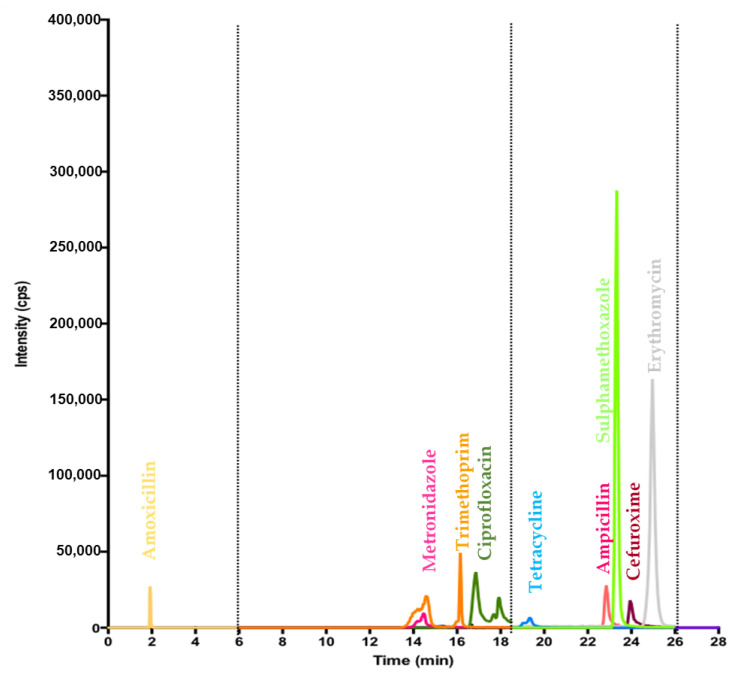
Chromatogram showing separated antibiotics and their peaks. The dotted vertical lines indicate 3 periods created during the analysis to improve upon the sensitivity and reliability of peak areas.

**Table 1 antibiotics-11-00119-t001:** Multiple drug resistance as observed by CoNS species.

Species	Community	Resistance to ≥4 Antibiotics (MDR CoNS)	Resistance to <4 Antibiotics (Non-MDR CoNS)	Total (%)
*S. haemolyticus*	Dodowa	24 (68.6)	4 (28.6)	28 (57.1)
	Korle-Gonno	11 (31.4)	10 (71.4)	21 (42.9)
	Sub-Total	35 (71.4)	14 (28.6)	49 (77.8)
*S. epidermidis*	Dodowa	3 (75.0)	1 (100.0)	4 (80.0)
	Korle-Gonno	1 (25.0)	0 (0.0)	1 (20.0)
	Sub-Total	4 (80.0)	1 (20.0)	5 (7.9)
Other CoNS *	Dodowa	3 (60.0)	2 (50.0)	5 (55.6)
	Korle-Gonno	2 (40.0)	2 (50.0)	4 (44.4)
	Sub-Total	5 (55.6)	4 (44.4)	9 (14.3)
	Total	44 (69.8)	19 (30.2)	63

* Other CoNS include *S. sciuri* (*n* = 2), *S. simulans *(*n* = 2), *S. hominis* (*n* = 2), *S. cohnii* (*n* = 1), *S. warnerii* (*n* = 1), *S. saprophyticus* (*n* = 1).

## Data Availability

The processed data presented in this study are included within this article and the Appendix A. The source data are not included due to ethical considerations but are however available on request from the corresponding author.

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
