# Peer review of "Reservoir of Antibiotic Residues and Resistant Coagulase Negative Staphylococci in a Healthy Population in the Greater Accra Region, Ghana"

_antibiotics, 2022, doi:10.3390/antibiotics11010119_

Round 1
Reviewer 1 Report
The manuscript entitled ‘Reservoir of Antibiotic Residues and Resistant Coagulase Negative Staphylococci in a Healthy Population in the Greater Accra Region, Ghana’ reported the occurrence of resistant Coagulase-Negative Staphylococci (rCoNS) and antibiotic residues in urine samples of 401 healthy individuals from Korle-Gonno (KG) and Dodowa (DDW) in Ghana. This work makes sense and can be considered to be published after the following question is addressed.
Comments:
- The LOD, LOQ, or screening limit of the antibiotics in the urine test should be specified in the results section.
- Ciprofloxacin is a metabolite of enrofloxacin. Its intended or unintended use or intake should be discussed in the discussion.
- Is there any other exposure source like water, drinking? The investigation of the dietary structure can be a good method for searching the source.
- The age and gender distribution of the individuals from which the urine samples were collected are suggested to be specified.
- The MALDI-TOF/MS method for identification of the CoNS should be well specified in terms of its accuracy, quality control, as it is a novel method different from the conventional biochemical approach.
- In the sample preparation and Clean up, the cartridge of solid-phase extract should be specified in terms of its type, and also, its performance should be demonstrated.
- What’s the column used for chromatographic separation? Please specify it.
- It seems like the peaks are not well separated, and the peak shape is not good enough? Figure 5 shows this.
- In the abstract, the residual concentration level should be summarized for the mentioned antibiotics.
Author Response
Authors’ reply to Review Report (Reviewer 1)
|
Sr No |
Comment |
Response |
|
1 |
The LOD, LOQ, or screening limit of the antibiotics in the urine test should be specified in the results section. |
The LOD, LOQ and the concentration range for the antibiotics in the validation has been included in the results under section 2.2. Refer to lines 128 to 132
This revision also resulted in the reorganization of the supplementary results and their references in the manuscript. |
|
2 |
Ciprofloxacin is a metabolite of enrofloxacin. Its intended or unintended use or intake should be discussed in the discussion. |
Although Enrofloxacin produces ciprofloxacin as an active metabolite, its use is not reported in Ghana. Ciprofloxacin is recommended in the Ghanaian Standard Treatment Guidelines for the management of urinary tract infections among other indications. Thus, ciprofloxacin formulations are widely marketed for use in Ghana. Hence its inclusion in the study. |
|
3 |
Is there any other exposure source like water, drinking? The investigation of the dietary structure can be a good method for searching the source. |
There has not been any report of antibiotic detection in potable water sources in Ghana. As such, its potential source as a route of antibiotic exposure was not considered in the scope of the study. |
|
4 |
The age and gender distribution of the individuals from which the urine samples were collected are suggested to be specified. |
The age and gender distribution of the individuals have been included under section 4.2. Lines 263 to 264. |
|
5 |
The MALDI-TOF/MS method for identification of the CoNS should be well specified in terms of its accuracy, quality control, as it is a novel method different from the conventional biochemical approach. |
The use of MALDI-TOF/MS method to identify CoNS and other microorganisms is not actually a new method as used in this study. The MALDI Biotyper (which was used in the current study and is one of the FDA-approved set-ups for microbial identification) (Line 281) determines the proteomic fingerprint of the CoNS organism and compares the characteristic patterns with an extensive reference fingerprint library of microorganisms to determine its identity.
The accuracy, sensitivity, specificity, and quality assurance of these set-ups are widely established and usually form part of consideration during regulatory approvals. Internally, however, these parameters are regularly monitored for routine use. This was ensured in the current study and has been indicated accordingly in Lines 282-283.
Below are some references provided in respect of MALDI-TOF applications in the identification of microorganisms:
· Rychert J. Benefits and limitations of MALDI-TOF mass spectrometry for the identification of microorganisms. Journal of Infectiology. 2019 Jul 2;2(4). · https://www.bruker.com/en/products-and-solutions/microbiology-and-diagnostics/microbial-identification.html · Sogawa K, Watanabe M, Sato K, Segawa S, Ishii C, Miyabe A, Murata S, Saito T, Nomura F. Use of the MALDI BioTyper system with MALDI–TOF mass spectrometry for rapid identification of microorganisms. Analytical and bioanalytical chemistry. 2011 Jun;400(7):1905-11. · El-Nemr IM, Mushtaha M, Sundararaju S, Fontejon C, Suleiman M, Tang P, Goktepe I, Hasan MR. Application of MALDI biotyper system for rapid identification of bacteria isolated from a fresh produce market. Current microbiology. 2019 Mar;76(3):290-6. |
|
6 |
In the sample preparation and Clean up, the cartridge of solid-phase extract should be specified in terms of its type, and also, its performance should be demonstrated. |
Details on the cartridges used are provided in Section 4.4.3 (Lines 318-320). The performance of the cartridges is described in lines 330-331 and its outcome is included in section 2.2 (Lines 132-133). |
|
7 |
What’s the column used for chromatographic separation? Please specify it |
The details on the columns used for the analysis have been included in section 4.4.4 (Lines 336-337) |
|
8 |
It seems like the peaks are not well separated, and the peak shape is not good enough? Figure 5 shows this. |
Generally, the peak shapes for the nine compounds were well resolved and separated with distinct retention times. The retention times have been included as an additional column in Table S9.
Also, the detection and quantitation of the antibiotics were based on the peak intensities of the characteristic quantifier to qualifier ratios of each analyte, and this was used to confirm the selectivity and specificity of the method. |
|
9 |
In the abstract, the residual concentration level should be summarized for the mentioned antibiotics. |
The range of concentrations has been included in the abstract. Line 35 |
Reviewer 2 Report
The manuscript entitled "Reservoir of Antibiotic Residues and Resistant Coagulase Negative Staphylococci in a Healthy Population in the Greater Accra Region, Ghana" by Samuel Oppong Bekoe and co-authors reported a study aiming at screening for the antibiotic-resistant coagulase-negative Staphylococci strains and the residues of antibiotics in healthy individuals in Ghana. The authors combined using analytical techniques and antimicrobial testing to isolate and characterize the CoNS strains as well as quantification of the antibiotic residues in the urine samples. The authors did a great job of accomplishing this manuscript, and the conclusions are also strongly supported by the data provided. Most importantly, the authors' concern about the widespread and improper use of antibiotics might cause severe problems for the management of infectious diseases in the hospital. And, this concern is reasonable since the authors did observe a strong correlation between the resistant CoNS and antibiotic residues in their study. I only have some minor points to add in order to make this article more understandable for a general audience.
- The authors might want to include more background about Coagulase Negative Staphylococci (CoNS) in the introduction. It is unclear why CoNS is considered harmless, and it is more confusing when the authors also mention it becomes a concern. Only rCoNS is a concern, or does it matter where the infections are, for example, on the skin or in the blood?
- The authors defined the MDR as expressing resistance to 4 or more antibiotics. However, in table 1, <4 is also labeled as MDR as stated "<4 (% of total #MDR CoNS)". That is just a little confusion.
- The authors need to include a detailed statistic calculation method in the Figure legends of Figures 1, 2, and 3, and also denote detailed p-value for the cut-off of statistical difference at * or ** (for example, * p<0.05, ** p< 0.001?).
- In Figure 4, the legend says, "*data were shown to be not normally distributed)". However, I failed to locate the * on the figure. Could the authors highlight this * marker?
- Still in Figure 4 legend, the authors need to include the full names of the abbreviations labeled in the figure for different antibiotics. Alternatively, the authors can use the full name, just like Figure 3.
Author Response
Authors’ reply to Review Report (Reviewer 2)
|
Sr No |
Comment |
Response |
|
1 |
The authors might want to include more background about Coagulase Negative Staphylococci (CoNS) in the introduction. It is unclear why CoNS is considered harmless, and it is more confusing when the authors also mention it becomes a concern. Only rCoNS is a concern, or does it matter where the infections are, for example, on the skin or in the blood? |
CoNS were reported to be commensal organisms, which were harmless when present on or in humans. However, other reports clearly indicate the virulent nature of these organisms. Thus, the presence of CoNS is no more harmless as thought previously. To further complicate the issue, resistant CoNS have also come up. Thus, in this study, we detected CoNS in general and amongst them, focus on the resistant ones observed.
Parts of the introduction has been revised to clarify the main objective of the study. Lines 64-76 |
|
2 |
The authors defined the MDR as expressing resistance to 4 or more antibiotics. However, in table 1, <4 is also labeled as MDR as stated "<4 (% of total #MDR CoNS)". That is just a little confusion. |
The confusion has been cleared. As per this study, MDR definition has been limited to only CoNS with resistance to 4 or more antibiotics. Refer to Table 1 (Line 89) |
|
3 |
The authors need to include a detailed statistic calculation method in the Figure legends of Figures 1, 2, and 3, and also denote detailed p-value for the cut-off of statistical difference at * or ** (for example, * p<0.05, ** p< 0.001?). |
Descriptions of the statistical tests used in the figures 1,2 & 3 have been added in their respective captions, including the cut-offs for statistical differences. |
|
4 |
In Figure 4, the legend says, "*data were shown to be not normally distributed)". However, I failed to locate the * on the figure. Could the authors highlight this * marker? |
The phrase ‘*data were shown to be not normally distributed was made in reference to the use of median in the data visualization. It has been taken off to avoid any ambiguity. Lines 162-163 |
|
5 |
Still in Figure 4 legend, the authors need to include the full names of the abbreviations labeled in the figure for different antibiotics. Alternatively, the authors can use the full name, just like Figure 3. |
Full names for the abbreviations have been provided in the caption for the figure. Lines 163 – 165. |
Round 2
Reviewer 1 Report
I believe it meets the standards for publication.